Local-global multi-scale attention network for medical image segmentation

http://orcid.org/0009-0000-3621-2142 Zhu Minghui 1
Cheng Dapeng 1 2 chengdapeng@sdtbu.edu.cn
Mao Yanyan 1
Sun Lu 3 sunnystone.happy@163.com
Jing Wanting 4
1 School of Computer Science and Technology, Shandong Technology and Business University , Yantai, Shandong , China
2 Yantai Key Laboratory of Big Data Modeling and Intelligent Computingy , Yantai, Shandong , China
3 Department of Oncology Jinan, Shandong Provincial Hospital Affiliated to Shandong First Medical University , Jinan, Shandong , China
4 School of Information and Electronic Engineering, Shandong Technology and Business University , Yantai, Shandong , China
Coelho Paulo Jorge
Electronic publication date: 2025 Jul 18
Publication date: 2025
Volume: 11
Electronic Location ID: e3033
Received 2025 Apr 17; Accepted 2025 Jun 24
Copyright: © 2025 Zhu et al.
Copyright year: 2025
Copyright holder: Zhu et al.
License: This is an open access article distributed under the terms of the Creative Commons Attribution License, which permits unrestricted use, distribution, reproduction and adaptation in any medium and for any purpose provided that it is properly attributed. For attribution, the original author(s), title, publication source (PeerJ Computer Science) and either DOI or URL of the article must be cited.
License URL: https://creativecommons.org/licenses/by/4.0/

Keywords: Efficient multi-scale attention, Local and global information extraction, Medical image segmentation

Funding: The authors received no funding for this work.

==============================
With the continuous advancement of deep learning technologies, deep learning-based medical image segmentation methods have achieved remarkable results. However, existing segmentation approaches still face several key challenges, including the insufficient extraction of local and global information from images and the inaccurate selection of core features. To address these challenges, this article proposes a novel medical image segmentation architecture—local-global multi-scale attention network (LGMANet). LGMANet introduces an innovative local-global information processing block (LGIPB) to effectively facilitate the deep mining of both local and global information during the downsampling process. In addition, an efficient multi-scale reconstruction attention (EMRA) module is designed to help the model accurately extract core features and multi-scale information while effectively suppressing irrelevant content. Experiments on the ISIC2018, CVC-ClinicDB, BUSI, and GLaS datasets demonstrate that LGMANet achieves IoU scores of 85.28%, 82.67%, 70.07%, and 88.90%, respectively, showcasing its superior segmentation performance.

Introduction

Medical image analysis plays a crucial role in disease diagnosis, treatment, and prognosis evaluation. Traditional methods primarily rely on manual interpretation, meaning that analysis results depend heavily on the doctor’s skill level and are subject to a degree of subjectivity, which can lead to inaccuracies (Chen et al., 2019). With the advancements in computer technology and deep learning, deep learning-based image processing and segmentation techniques (Liu et al., 2024; Wang et al., 2024a; Li et al., 2025; Wang et al., 2024b; Xing et al., 2022; Li et al., 2024b; Xing et al., 2024; Shaker et al., 2024), have begun to be widely used. At the same time, computer-aided diagnostic technologies have been continuously developed, significantly improving the efficiency and accuracy of medical professionals while also reducing treatment costs.

In 2015, Ronneberger, Fischer & Brox (2015) first proposed U-Net, a deep learning based neural network for medical image segmentation, which enables end-to-end training with minimal data while producing high-quality segmentation results. Since then, researchers have continuously optimized network architectures by introducing various enhancements. Some models have incorporated attention mechanisms to focus on target regions, thereby improving segmentation accuracy (Oktay et al., 2018). Others have utilized modules such as dilated convolutions to expand the receptive field and boost learning capacity (Chen et al., 2017). Additionally, enhancements to the U-Net architecture have been made by optimizing its encoder and decoder components to further improve segmentation performance (Isensee et al., 2021; Zhou et al., 2019). With the advent of Transformers, integrating convolutional neural networks (CNNs) with Transformer architectures has been explored to enhance the model’s ability to capture complex image structures, thereby increasing segmentation accuracy (Chen et al., 2021; Cao et al., 2022). In recent years, studies have also focused on incorporating few-shot learning to address the challenge of limited annotated data in medical imaging.

Although current deep learning based medical image segmentation methods have shown promising results, they continue to face several critical challenges. Chief among these is the insufficient extraction of both local and global information from medical images (Cheng et al., 2024a; Ruan et al., 2023; Cheng et al., 2024b, 2023). Additionally, the identification of essential image features is often inaccurate (Zhang et al., 2025; Yuan, Song & Fan, 2024; Ruan et al., 2023). Together, these limitations significantly hinder the model’s ability to learn effectively, ultimately compromising segmentation accuracy.

To address these challenges, this article proposes a novel local-global multi-scale attention network (LGMANet). Extensive experiments conducted on various medical image datasets demonstrate that LGMANet outperforms existing state-of-the-art methods across multiple evaluation metrics. The main contributions of this article are as follows: We propose a novel local-global multi-scale attention network (LGMANet) to address the challenges in current medical image segmentation tasks, including the insufficient extraction of both local and global information, as well as the inaccurate identification of core image features.

LGMANet incorporates two key components: an innovative local-global information processing block (LGIPB) and an efficient multi-scale reconstruction attention (EMRA) module. The LGIPB is designed to fully extract both local and global information during the image downsampling process, while the EMRA module accurately captures core image features, effectively suppresses irrelevant information, and enhances the model’s ability to learn multi-scale features.

Extensive experimental results across four datasets-ISIC2018, BUSI, CVC-ClinicDB, and GLaS based on multiple evaluation metrics demonstrate that the proposed LGMANet outperforms existing state-of-the-art medical image segmentation methods.

In the following sections, this article discusses the design and implementation of the proposed method. “Related Work” reviews related work. “Method” provides a detailed explanation of the proposed method and its loss function. “Experiments” presents the experimental setup, dataset usage, and results. Finally, conclusions are provided in “Conclusion”.

Related work

This section reviews the related work pertinent to the research presented in this article. The review focuses on traditional medical image segmentation methods, deep learning based approaches, and the key challenges currently faced in medical image segmentation tasks.

Traditional medical image segmentation methods

Traditional medical image segmentation methods include edge-based methods (Ding & Goshtasby, 2001), threshold-based methods (Dong, 2014), region-based methods (Lalaoui & Mohamadi, 2013; Freixenet et al., 2002), feature-based methods (Likas, Vlassis & Verbeek, 2003), and active contour-based algorithms (Park, Schoepflin & Kim, 2001). Edge-based methods identify image boundaries by detecting changes in image gradients, but they are susceptible to noise and complex backgrounds. Threshold-based methods perform image segmentation by utilizing gray-level histograms to separate the foreground from the background, but they perform poorly on images with complex gray-level distributions. Region-based methods segment images by merging or splitting similar pixels, yet they are highly sensitive to noise. Feature-based methods perform segmentation using texture, color, and other features, but they heavily rely on handcrafted feature design and the quality of training data. Active contour-based methods are capable of handling complex shapes, but they typically involve high computational costs. Overall, these methods are relatively inefficient and lack stability when handling complex data. In recent years, deep learning-based approaches have become a major research focus (Ottesen et al., 2023; Umirzakova et al., 2024).

Deep learning-based medical image segmentation methods

With the rapid development of deep learning technology, an increasing number of researchers have begun to adopt deep learning methods to tackle medical image segmentation tasks. This section will introduce medical image segmentation methods based on convolutional neural networks and those based on attention mechanisms, respectively.

Convolution-based methods

In 2015, Ronneberger, Fischer & Brox (2015) improved the fully convolutional network (FCN) and proposed U-Net, a deep neural network specifically designed for medical image segmentation tasks. This network introduced a symmetric encoder-decoder architecture and utilized skip connections to combine low-level and high-level features, enabling efficient training and producing high-quality segmentation results. In 2016, Kamnitsas et al. (2016) proposed DeepMedic, a multi-channel deep convolutional neural network focused on the semantic segmentation of medical images. DeepMedic combines local and global convolutions, enabling precise tumor region segmentation. Milletari, Navab & Ahmadi (2016) introduced V-Net, a CNN designed for 3D medical image segmentation. This model employs 3D convolutions and introduces the Dice loss function, effectively handling volumetric medical images. In 2018, Zhang, Liu & Wang (2018) improved the U-Net model by proposing ResUNet, incorporating residual connections that mitigate the vanishing gradient problem in deep networks and help the network capture higher-quality features. Chen et al. (2018) proposed DeepLabV3+, an efficient image segmentation model based on atrous convolution and atrous spatial pyramid pooling (ASPP), which has achieved significant success in the field of medical image segmentation. Li et al. (2018) introduced DenseNet into the U-Net architecture and proposed DenseUNet, which utilizes dense connections to enhance feature reuse. This model improves feature representation capabilities while significantly reducing the number of parameters, making it particularly suitable for small-sample datasets in medical imaging. In 2019, Guo et al. (2019) proposed a supervised multimodal image analysis framework, integrating cross-modal fusion at three levels: feature learning, classifier, and decision-making.

Attention-based methods

In 2020, Jin et al. (2020) proposed RA-UNet, a deep learning network for automatic liver and tumor segmentation. This method combines dual-channel feature extraction and attention mechanisms to accurately extract the liver region and segment tumors, thereby improving segmentation accuracy. In 2021, Gao, Zhou & Metaxas (2021) proposed UTNet, a simple yet powerful hybrid Transformer architecture designed to enhance medical image segmentation by integrating the self-attention mechanism into convolutional neural networks. In 2022, Hatamizadeh et al. (2022) proposed UNETR (UNET Transformers), an architecture that employs a transformer as the encoder to learn sequence representations of the input volume, effectively capturing global multi-scale information while retaining the successful “U-shaped” network design for both the encoder and decoder. In 2023, Ruan et al. (2023) proposed Efficient Group Enhanced UNet (EGE-UNet), which integrates a Group multi-axis Hadamard Product Attention module (GHPA) and a group aggregation bridge module (GAB) in a lightweight manner, achieving excellent performance. In 2024, Li et al. (2024a) introduced the Dynamic Spatial Group Enhanced Network (DSEUNet), which significantly reduces computational load and the number of parameters while maintaining competitive segmentation accuracy.

Issues that need to be addressed

Despite significant advancements in existing medical image segmentation methods, several issues still remain: 1. Current deep learning based medical image segmentation methods still face the challenge of insufficient extraction of both local and global information. This limitation prevents the networks from fully capturing the detailed features of the images, ultimately affecting the accuracy of the segmentation results.

2. Many existing methods struggle with inaccuracies in selecting core image features, failing to effectively identify and emphasize the most critical regions of the image. As a result, this undermines the model’s learning capacity and reduces the precision of the segmentation outcomes.

To address these challenges, we propose a novel medical image segmentation method-local-global multi-scale attention network (LGMANet). Compared to traditional medical image segmentation methods, LGMANet effectively overcomes issues such as reliance on handcrafted feature extraction, susceptibility to noise, and reduced accuracy when handling images with complex backgrounds. In contrast to existing deep learning based medical image segmentation methods (Umirzakova et al., 2024), LGMANet integrates the local-global information processing blocks (LGIPBs) and the efficient multi-scale reconstruction attention (EMRA) modules, enabling the deep mining of both local and global information from images and more precise selection of core features. This effectively addresses the challenges of insufficient local-global information extraction and inaccurate core feature selection in existing methods.

Method

This section provides a comprehensive overview of LGMANet’s architecture and its loss function, focusing on its innovative key components: the LGIPB and the ERMA module.

Overall structure

To address the main challenges in current medical image segmentation tasks, such as the insufficient extraction of both local and global information from images and the inaccurate selection of core image features, we designed the LGMANet. The specific structure of LGMANet is illustrated in Fig. 1. Inspired by the U-Net model (Ronneberger, Fischer & Brox, 2015), LGMANet adopts an effective skip connection structure and a five-layer encoder-decoder framework as its basic architecture. In the encoder, the first three stages of LGMANet each consist of two innovative LGIPBs, replacing the standard convolutional layers, thereby deeply mining and extracting both local and global information during the image downsampling process. The subsequent two stages employ shifted MLP modules to better capture complex feature relationships, further enhancing the overall expressive power and generalization capabilities. In the decoder, the first stage employs the shifted MLP module, the second stage integrates the shifted MLP with the EMRA module, and the remaining three stages combine the EMRA module with convolutional layers. This decoder structure accurately extracts core features, effectively suppresses irrelevant information, and has strong capabilities for learning multi-scale image information.

Figure 1 LGMANet framework: incorporating the innovative LGIPB and EMRA modules to address challenges in local-global information extraction and core feature identification for medical image segmentation.

We first briefly introduce the shifted MLP, which is a key component of our network architecture. Given the remarkable performance of shifted MLP (Liu et al., 2021) in medical image segmentation and related tasks, this study adopts its architecture to enhance the network’s ability to model both local and global feature dependencies. Shifted MLP offers several significant advantages, including its dynamic shifting mechanism, which captures spatial dependencies across neighboring regions, the integration of depthwise convolution to further enhance local feature extraction, and its efficient modeling of global features through linear transformations. This combination achieves a balance between computational efficiency and representational power. The shifted MLP module consists of several key components. First, a linear layer projects the input features into a higher-dimensional feature space. Next, a dynamic shifting operation, combined with depthwise convolution, captures both local and long-range feature dependencies. Finally, another linear layer maps the features back to their original dimensionality. The module also incorporates activation functions (e.g., GELU) and dropout mechanisms to enhance non-linear expressiveness and mitigate overfitting. This module was first proposed by Liu et al. (2021), with further implementation details available in their original work.

Local-global information processing block

To address the challenge of insufficient extraction of both local and global information in medical images, this article introduces a LGIPB, whose detailed structure is shown in Fig. 2. It is important to note that when LGIPB is positioned at the beginning of the network encoder, its input is a pathological image; otherwise, the input is a feature map. Additionally, the output of LGIPB is always a feature map. Specifically, the input feature map (or pathological image) is first processed by a 3×3 convolutional layer to preliminarily extract image information. Subsequently, the processed feature map undergoes three 3×3 convolutional layers with dense connectivity structures, each equipped with batch normalization and Prelu activation functions, to effectively fuse information from different levels as well as local and global features. Finally, the output feature map is processed by another 3×3 convolutional layer with batch normalization and Prelu activation functions to adjust the number of channels back to that of the input channels. Parametric Rectified Linear Unit (PReLU) is a nonlinear activation function whose slope in the negative axis is determined by a parameter that is automatically learned during training through backpropagation. This parameter is initialized to a small positive value (this model adopts the widely used default value of 0.25) and is continuously adjusted during training. The purpose of this parameter is to prevent the “dying neuron” problem and to make the model’s response in the negative region more flexible, thereby enhancing the model’s representation capability and generalization performance.

Figure 2 The illustration of LGIPB: enhancing the model’s local-global information extraction through dense connection structures.

This module effectively fuses feature information from various levels by progressively increasing the number of channels and concatenating features, thereby enhancing the model’s ability to represent complex features and capture both local and global information. Additionally, the dense connectivity structure within the module promotes feature reuse and information flow, alleviates the vanishing gradient problem, and improves the model’s training efficiency and overall performance. This design not only enhances the model’s performance in medical image segmentation tasks but also strengthens its ability to capture multiscale features, enabling the model to better handle medical images with complex structures and diverse characteristics.

To facilitate readers’ understanding, the pseudocode for the LGIPB is provided in Algorithm 1. In the pseudocode, cat(⋅) represents the feature concatenation operation, which concatenates multiple feature maps along the channel dimension; Conv3×3(⋅) denotes a 3×3 convolution operation; BatchNorm(⋅) indicates batch normalization, which standardizes the feature maps to accelerate model convergence and enhance stability; and Prelu(⋅) refers to a parametric rectified linear activation function that introduces nonlinearity.

Algorithm 1 Algorithm for LGIPB.

 1: Input: xin	
 2: Output: xout	
 3:  xconv1←Conv3×3(xin)))	
 4: for i=1 to 3 do	
 5:   if i=1 then	
 6:      xconvi+1←Prelu(BatchNorm(Conv3×3(xconvi)))	
 7:   else	
 8:      xconvi+1←Prelu(BatchNorm(Conv3×3(cat(xconvi,xconvi−1...,xconv1))))	
 9:   end if	
10: end for	
11:  xout←Prelu(BatchNorm(Conv3×3(xconv4)))	
12: return xout	

For a more rigorous mathematical perspective on the detailed structure of the LGIPB and to more clearly illustrate its internal mechanisms, we also present a set of equations describing the LGIPB. Let the input feature map be xin∈RB×Cin×H×W and the output feature map be xout∈RB×Cout×H×W. Here, B represents the batch size, Cin represents the number of input channels, Cout represents the number of output channels, H represents the image height, and W represents the image width. The main operations are described by Eqs. (1)–(5):

(1) x1=Conv3×3(xin),

(2) x2=Prelu(BatchNorm(Conv3×3(x1))),

(3) x3=Prelu(BatchNorm(Conv3×3(cat(x1,x2)))),

(4) x4=Prelu(BatchNorm(Conv3×3(cat(x1,x2,x3)))),

(5) xout=Prelu(BatchNorm(Conv3×3(x4))).

Here, the meanings of cat(⋅), Conv3×3(⋅), BatchNorm(⋅), and Prelu(⋅) remain the same as in Algorithm 1. Specifically, in Eq. (1), a 3×3 convolution maps the input xin from Cin channels to 12 channels, producing x1. Next, in Eq. (2), a second 3×3 convolution followed by batch normalization and Prelu activation yields x2 (12 channels). In Eq. (3), x1 and x2 are concatenated and fed into another 3×3 convolution, batch normalization, and Prelu, resulting in x3 (24 channels). Then, in Eq. (4), x1,x2,x3 are concatenated and passed through a 3×3 convolution, batch normalization, and Prelu, generating x4 (48 channels). Finally, in Eq. (5), a 3×3 convolution maps 48 channels to Cout, followed by batch normalization and Prelu, yielding the output xout.

Efficient multi-scale reconstruction attention

To address the issue of insufficient precision in selecting core features of medical images, we have designed an EMRA module. This module is inspired by ECANet (Wang et al., 2020), and its detailed structure is shown in Fig. 3.

Figure 3 The illustration of EMRA module: enhancing the model’s feature selection and multi-scale information extraction with adaptive attention.

Specifically, in the EMRA module, the input feature map is first processed using an attention mechanism. The attention mechanism starts with global average pooling to capture the global statistical features of each channel. Next, a one-dimensional convolution with an adaptive kernel size, followed by a sigmoid activation function, computes the channel attention weights. These weights represent the importance of each channel: higher weights correspond to features that contribute more to the final task, while lower weights indicate redundant or irrelevant features. Finally, the attention weights are element-wise multiplied with the original input, suppressing redundant or irrelevant features and amplifying the critical ones. Next, the weighted feature map is fed into multi-scale convolutional layers with varying kernel sizes (1 × 1, 3 × 3, 5 × 5, and 7 × 7) to capture features across different receptive fields. These features are then concatenated along the channel dimension to enhance their representational capacity. Finally, a 1 × 1 convolution compresses the concatenated feature map back to the target number of channels, producing the final output feature map. Through this process, the EMRA module effectively extracts multi-scale information from the image, highlighting key features while suppressing redundant or noisy information, thereby improving the model’s adaptability to complex image patterns. By integrating the attention mechanism with multi-scale feature fusion, the network achieves more accurate selection of important features in medical images, significantly enhancing its robustness and representational power.

To better illustrate its functionality, the pseudocode of the EMRA module is provided in Algorithm 2. In this pseudocode, AvgPool(⋅) denotes the global average pooling operation, which generates channel descriptors, while Reshape(⋅,[B,1,C]) and Reshape(⋅,[B,C,1,1]) represent tensor reshaping operations to adjust the feature map to the required shapes for subsequent processing. Conv1D(⋅) indicates the 1D convolution operation used to model interactions between channels, ⊙ denotes element-wise multiplication, and Sigmoid(⋅) represents the Sigmoid activation function for normalizing attention weights. The Expand(⋅) operation extends the attention weights to match the size of the input feature map for element-wise weighting. Additionally, Conv1×1(⋅), Conv3×3(⋅), Conv5×5(⋅), and Conv7×7(⋅) correspond to convolution operations with kernel sizes of 1×1, 3×3, 5×5, and 7×7, respectively, to extract features at different receptive fields. Finally, cat(⋅) denotes the feature concatenation operation, which integrates multi-scale features along the channel dimension into a unified representation.

Algorithm 2 Algorithm for EMRA.

 1: Input: xin	
 2: Output: xout	
 3:  y1←AvgPool(xin)	
 4:  y2←Reshape(y1,[B,1,C])	
 5:  y3←Conv1D(y2)	
 6:  y4←Sigmoid(y3)	
 7:  y5←Reshape(y4,[B,C,1,1])	
 8:  xweighted←xin⊙Expand(y5)	
 9:  feat1←Conv1×1(xweighted)	
10:  feat3←Conv3×3(xweighted)	
11:  feat5←Conv5×5(xweighted)	
12:  feat7←Conv7×7(xweighted)	
13:  fused←cat(feat1,feat3,feat5,feat7)	
14:  xout←Conv1×1(fused)	
15: return xout	

For a more rigorous mathematical perspective on the detailed structure of the EMRA module and to clearly illustrate its internal mechanisms, we provide a detailed mathematical description. Let c denote the number of channels in the input tensor, and let γ and b be tunable parameters. In the implementation, the 1D convolution kernel size k is first determined by Eq. (6).

(6) t=⌊|log2⁡(c)+b|γ⌋⟹k={t,iftmod2=1,t+1,iftmod2=0,

where log2(⋅) denotes the base-2 logarithm, and ⌊⋅⌋ is the floor operation. Ensuring k is odd helps maintain a symmetric channel interaction for the 1D convolution. In the EMRA module, let the input feature map be xin∈RB×Cin×H×W and the output feature map be xout∈RB×Cout×H×W. Here, B represents the batch size, Cin represents the number of input channels, Cout represents the number of output channels, H represents the image height, and W represents the image width. The main operations are described by Eqs. (7)–(9):

(7) xweighted=xin⊙Expand(Reshape(Sigmoid(Conv1Dk(Reshape(AvgPool(xin),[B,1,C]))),[B,C,1,1])),

(8) f1=Conv1×1(xweighted),f3=Conv3×3(xweighted),f5=Conv5×5(xweighted),f7=Conv7×7(xweighted)

(9) xout=Conv1×1(cat(f1,f3,f5,f7)).

where Conv1Dk denotes a 1D convolution with kernel size k (from Eq. (6)), the meanings of AvgPool(⋅), Reshape(⋅,[B,1,C]), Reshape(⋅,[B,C,1,1]), ⊙, Sigmoid(⋅), Expand(⋅), Conv1×1(⋅), Conv3×3(⋅), Conv5×5(⋅), Conv7×7(⋅) and cat(⋅) remain the same as in Algorithm 2. Specifically, in Eq. (7), the attention mechanism begins by applying global average pooling to xin, reshaping the result, processing it with a 1D convolution of kernel size k, and applying a Sigmoid activation. The generated attention weights are then used to reweight the input feature map, yielding the weighted feature map xweighted∈RB×Cin×H×W. At this stage, the channel dimension remains Cin.

Subsequently, in Eq. (8), xweighted is fed into four parallel convolutions with kernel sizes 1×1, 3×3, 5×5, and 7×7, producing multi-scale feature maps f1,f3,f5,f7∈RB×Cin×H×W. Here, B represents the batch size, Cin represents the number of input channels, H represents the image height, and W represents the image width. Each fi preserves the channel count Cin. Finally, in Eq. (9), these feature maps are concatenated along the channel dimension, resulting in a feature map fconcat∈RB×4Cin×H×W, and passed through a 1×1 convolution to reduce the channels to Cout, producing the output xout∈RB×Cout×H×W.

Dice loss

In segmentation tasks, the imbalance between background and foreground pixels is a common challenge. To address this issue, the Dice coefficient loss (DL) is widely used, as it effectively alleviates the impact of pixel class imbalance and significantly improves segmentation performance. Therefore, this study also employs this loss function.

The DL modifies the segmentation evaluation metric, the Dice similarity coefficient (DSC), to optimize the consistency between predicted samples and ground truth annotations. It has demonstrated superior performance in segmentation tasks. The formula is shown in Eq. (10):

(10) DL(p,g)=1−2∑i=1Npigi+ε∑i=1Npi2+∑i=1Ngi2+ε

where ε, ranging from [0,1], is a tunable parameter designed to prevent division by zero errors and improve gradient propagation for negative samples.

It is worth noting that, in this study, we employ a single loss function only at the final output stage, rather than assigning and weighting separate loss functions at each upsampling stage, as is typical in U-Net architectures. The main reasons for this choice are as follows: on the one hand, the introduction of the LGIPBs and the EMRA modules in our model (compared to a standard U-Net design) increases network complexity to some extent. If we also assigned and weighted multiple loss functions at each upsampling stage, the model structure would become excessively complicated, slowing down training and potentially leading to instability. In contrast, using a single loss function at the final output stage allows the model to focus on the quality of the final segmentation result, avoiding interference from intermediate feature map optimization. This approach reduces the competition among gradients at different resolutions and stabilizes the training process. On the other hand, our experimental results show that integrating the LGIPBs and the EMRA modules significantly enhances the network’s learning ability, making it feasible to achieve favorable performance with just a single loss function at the final output stage.

Experiments

Experimental datasets

This study evaluates the efficacy of LGMANet using four widely used, publicly available biomedical imaging datasets: GlaS (Sirinukunwattana et al., 2017), ISIC2018 (Codella et al., 2019; Tschandl, Rosendahl & Kittler, 2018) BUSI (Al-Dhabyani et al., 2020), and CVC-ClinicDB (Bernal et al., 2015). These datasets include both the original images and their corresponding mask images, and they are commonly employed in medical image segmentation tasks. It should be noted that the annotations in these datasets focus primarily on lesion areas, without separate labeling of normal tissues, because the main goal of medical image segmentation is to assist clinical diagnosis and treatment by accurately identifying the location and shape of lesion regions, while normal tissues are usually regarded as background.

For the ISIC2018 dataset (Codella et al., 2019; Tschandl, Rosendahl & Kittler, 2018), the images are resized to 512×512 for both training and testing. For the remaining three datasets, the images are resized to 384×384. The training-testing ratio follows standard practices in medical image segmentation, with 80% of the images allocated for training and the remaining 20% for model evaluation. Prior to being input into the model, the training data undergoe random rotations and random horizontal or vertical flips, along with normalization, to enhance data diversity and prevent instability due to varying input feature scales. More detailed information about the datasets can be found in Table 1.

Table 1 Medical datasets used in our experiments.

Dataset	Modality	Images	Input resolution	Train/Test	
CVC-ClinicDB	Endoscopy	612	384×384	490/122	
ISIC2018	Dermoscopy	2,594	512×512	2,075/519	
BUSI	Ultrasound	210	384×384	168/42	
GlaS	H&E stained sections	165	384×384	132/33	

Implementation details

All experiments in this article were conducted on an NVIDIA RTX 3060 GPU with 12GB of VRAM. The total number of training epochs was set to 400, with a batch size of 8. During training, the model employed the Adam optimizer and utilized a cosine annealing strategy for learning rate decay. The initial learning rate was set to 0.0001.

Experimental results

This subsection presents the numerical results of various methods evaluated on the four datasets used in this study. The evaluation metrics include Dice coefficient, intersection over union (IoU), recall, and precision, and the image outputs generated by each method are also compared. In addition, statistical significance tests were conducted between our method and several selected methods. To ensure fairness and accuracy, we reproduced these methods on the datasets used in this study based on open-source code, keeping the experimental parameters consistent with those used in the original articles.

Statistical significance test

To verify the robustness of performance differences between different models, this subsection presents a P-value analysis of the differences in IoU results between the proposed method and various other methods. The results are shown in Table 2. Specifically, we randomly selected 30 images from each of the ISIC 2018, CVC-ClinicDB, and BUSI datasets, performed segmentation using different methods, calculated the IoU of the segmentation results, and conducted paired t-tests against the IoU results generated by LGMANet to obtain the P-values.

Table 2 P-value analysis of IoU results between LGMANet and different methods across various datasets.

Dataset	UNet	MedT	Unext	EGE-UNet	
ISIC2018	0.011091	0.022373	0.009751	0.011617	
CVC-ClinicDB	0.000800	0.002177	0.000500	0.005172	
BUSI	0.000014	0.000056	0.039351	0.037879	

From the P-value results shown in the Table 2, it can be observed that all P-values are significantly smaller than the significance level of 0.05. This indicates that, from a statistical perspective, the proposed LGMANet exhibits significant performance differences in segmentation compared to the four mainstream methods (UNet, MedT, UNeXt, and EGE-UNet) across the three datasets (ISIC 2018, CVC-ClinicDB, and BUSI).

Quantitative results

The numerical results of LGMANet compared to other methods are presented in Tables 3–6. Compared to current state-of-the-art methods, LGMANet demonstrates superior performance across most metrics on all four datasets. Specifically, on the ISIC2018 dataset, LGMANet outperforms the second-best method in the Dice, IoU, and Recall metrics, achieving scores of 91.95%, 85.28%, and 93.02%, respectively, representing improvements of 0.6%,0.92%, and 2.06%. On the CVC-ClinicDB dataset, LGMANet also outperforms the second-best method in the Dice, IoU, and recall metrics, with scores of 90.45%, 82.67%, and 89.70%, respectively, showing improvements of 3.47%, 5.63%, and 6.12%. On the BUSI dataset, LGMANet achieves the highest performance in the Dice, IoU, and Precision metrics, with scores of 81.93%, 70.07%, and 79.45%, respectively, surpassing the second-best methods by 4.52%, 6.65%, and 0.06%. Finally, on the GlaS dataset, LGMANet leads in the Dice and IoU metrics, with scores of 94.11% and 88.90%, respectively, outperforming the second-best method by 0.59% and 0.99%.

Table 3 Comparison of various methods on the ISIC 2018 dataset.

Method	Dice	IoU	Recall	Precision	Parameters	
UNet (Ronneberger, Fischer & Brox, 2015)	85.14	75.01	80.71	91.55	31.04 M	
ResUNet (Zhang, Liu & Wang, 2018)	86.48	76.91	82.86	91.45	31.56 M	
Attention UNet (Oktay et al., 2018)	90.72	83.46	90.68	91.13	34.88 M	
MedT (Valanarasu et al., 2021)	88.27	79.70	85.72	91.66	1.56 M	
UNext (Valanarasu & Patel, 2022)	90.21	82.84	89.83	91.24	1.47 M	
MALUNet (Ruan et al., 2022)	88.83	80.57	90.33	88.09	0.175 M	
EGE-UNet (Ruan et al., 2023)	89.22	81.21	87.45	91.76	0.053 M	
EAV-UNet (Cheng et al., 2023)	86.93	77.46	84.96	90.02	32.67 M	
LeViT-UNet (Xu et al., 2023)	87.34	79.67	89.14	87.62	4.53 M	
AMNNet (Cheng et al., 2024a)	91.35	84.36	90.96	92.14	1.64 M	
RMIS-Net (Zhang et al., 2025)	90.45	83.81	90.39	90.16	5.47 M	
Ours	91.95	85.28	93.02	91.23	5.51 M	
Note:

The bold text in the “Method” column indicates the method proposed in this article, while the bold numbers under different metric columns represent the best performance scores.

Table 4 Comparison of various methods on the CVC-ClinicDB dataset.

Method	Dice	IoU	Recall	Precision	Parameters	
UNet (Ronneberger, Fischer & Brox, 2015)	82.71	71.17	79.44	87.25	31.04 M	
ResUNet (Zhang, Liu & Wang, 2018)	86.19	76.08	81.98	91.91	31.56 M	
Attention UNet (Oktay et al., 2018)	83.59	72.58	79.07	89.76	34.88 M	
MedT (Valanarasu et al., 2021)	67.38	58.88	60.39	82.67	1.56 M	
UNext (Valanarasu & Patel, 2022)	86.71	77.04	83.58	90.82	1.47 M	
MALUNet (Ruan et al., 2022)	71.78	56.45	69.29	76.10	0.175 M	
EGE-UNet (Ruan et al., 2023)	63.95	47.35	65.22	64.58	0.053 M	
EAV-UNet (Cheng et al., 2023)	85.39	74.95	80.21	92.25	32.67 M	
Ours	90.45	82.67	89.70	91.42	5.51 M	
Note:

The bold text in the “Method” column indicates the method proposed in this article, while the bold numbers under different metric columns represent the best performance scores.

Table 5 Comparison of various methods on the BUSI dataset.

Method	Dice	IoU	Recall	Precision	Parameters	
UNet (Ronneberger, Fischer & Brox, 2015)	71.96	56.39	69.92	75.60	31.04 M	
ResUNet (Zhang, Liu & Wang, 2018)	74.37	59.48	70.93	78.75	31.56 M	
Attention UNet (Oktay et al., 2018)	74.10	59.26	70.49	79.39	34.88 M	
MedT (Valanarasu et al., 2021)	53.12	39.45	54.95	61.96	1.56 M	
UNext (Valanarasu & Patel, 2022)	77.41	63.42	78.88	77.17	1.47 M	
MALUNet (Ruan et al., 2022)	67.77	51.84	66.18	72.00	0.175 M	
EGE-UNet (Ruan et al., 2023)	65.86	49.41	70.41	64.73	0.053 M	
EAV-UNet (Cheng et al., 2023)	73.82	59.23	86.42	62.72	32.67 M	
Ours	81.93	70.07	85.09	79.45	5.51 M	
Note:

The bold text in the “Method” column indicates the method proposed in this article, while the bold numbers under different metric columns represent the best performance scores.

Table 6 Comparison of various methods on the GlaS dataset.

Method	Dice	IoU	Recall	Precision	Parameters	
UNet (Ronneberger, Fischer & Brox, 2015)	93.40	87.69	93.52	93.34	31.04 M	
ResUNet (Zhang, Liu & Wang, 2018)	93.52	87.91	93.63	93.50	31.56 M	
Attention UNet (Oktay et al., 2018)	93.27	87.49	93.06	93.52	34.88 M	
MedT (Valanarasu et al., 2021)	78.91	66.80	89.63	73.34	1.56 M	
UNext (Valanarasu & Patel, 2022)	93.24	87.47	93.81	92.68	1.47 M	
MALUNet (Ruan et al., 2022)	84.28	73.48	79.32	90.76	0.175 M	
EGE-UNet (Ruan et al., 2023)	87.85	78.53	91.16	84.85	0.053 M	
EAV-UNet (Cheng et al., 2023)	93.30	87.54	95.65	91.09	32.67 M	
Ours	94.11	88.90	95.13	93.15	5.51 M	
Note:

The bold text in the “Method” column indicates the method proposed in this article, while the bold numbers under different metric columns represent the best performance scores.

It is worth noting that some methods perform well on certain metrics but relatively poorly on others. For example, on the GlaS dataset, EAV-UNet achieves a higher recall than our proposed method; however, its Dice and IoU scores are significantly lower. We believe this discrepancy may be due to EAV-UNet’s tendency to over-segment the target regions, which increases the recall but also introduces more false-positive areas. As a result, the overall overlap (Dice) and IoU decrease. This observation suggests that improving a single metric alone is not sufficient to fully reflect a model’s performance in medical image segmentation tasks. A better balance between precision and recall is essential. In contrast, LGMANet achieves strong performance across multiple key metrics, demonstrating its robustness and generalization ability in diverse scenarios.

In addition to model performance, we also compared the number of parameters across different methods. As shown in Tables 3–6, the number of parameters in our proposed LGMANet is at an acceptable level of 5.51M. Compared with methods such as UNet and EAV-UNet, it shows a significant reduction. The main reason is that LGMANet replaces the multi-layer convolutional structures in the last two layers of the encoder and the first two layers of the decoder in the traditional UNet architecture with shifted MLPs, which have significantly fewer parameters.

Qualitative results

The qualitative results of LGMANet and other methods are shown in Figs. 4–7. For the ISIC2018 dataset (Codella et al., 2019; Tschandl, Rosendahl & Kittler, 2018), five images were selected (each row represents a different image) for display, comparing them with current state-of-the-art methods. For the other three datasets, three images were selected (each row represents a different image) and compared with existing methods. In all displayed images, the first column on the left shows the original medical image input to the model, the second column presents the corresponding mask label provided in the dataset, and from the third column onward, each column shows the segmentation results of a different method. It can be observed that LGMANet achieves superior segmentation results compared to current state-of-the-art methods across most of the four datasets.

Figure 4 Segmentation results of different methods on the ISIC 2018 dataset.

As shown in the figure, LGMANet produces more accurate and stable segmentation structures. In contrast, EGE-UNet performs poorly on the images in the second and fourth rows, exhibiting significant missegmentation. MALUNet also shows noticeable errors in edge segmentation for the images in rows 2 to 4.

Figure 5 Segmentation results of different methods on the CVC-ClinicDB dataset.

As observed from the figure, LGMANet demonstrates consistently strong performance overall, while other methods such as AttentionUNet and MedT exhibit misclassification in certain normal regions.

Figure 6 Segmentation results of different methods on the BUSI dataset.

From the figure it can be seen that LGMANet makes some errors in segmenting the images in the second row but performs well on the other images. In contrast, methods such as EVA-UNet and MedT exhibit segmentation errors on multiple images.

Figure 7 Segmentation results of different methods on the GlaS dataset.

From the figure, it can be observed that LGMANet performs well in segmenting all three example images, whereas methods such as MedT and EGE-Unet exhibit some segmentation errors.

In addition to presenting conventional qualitative results, this article also demonstrates the effectiveness of LGMANet on specialized data. We showcase the segmentation results of various models on data containing noise interference, unclear boundaries, and overlapping regions, selected from four datasets, as shown in Figs. 8–11. Specifically, three images from each dataset were selected for display (each row represents a different image). In all the displayed images, the first column on the left shows the original medical image input to the model, the second column presents the corresponding mask label provided in the dataset, and from the third column onward, each column shows the segmentation results of a different method. Figure 8 shows images from the ISIC2018 dataset with noise interference (including hair and patches). Figures 9 and 10 display images with unclear boundaries from the CVC-ClinicDB and BUSI datasets, respectively. Figure 11 presents images from the GlaS dataset, featuring overlapping regions and unclear boundaries.

Figure 8 Segmentation results of different methods on the ISIC 2018 dataset: images affected by noise.

From the figure, it can be seen that LGMANet is able to accurately segment lesion areas even when the images are affected by noise. Additionally, its segmentation of edge regions is more precise compared to methods like ResUNet.

Figure 9 Segmentation results of different methods on the CVC-ClinicDB dataset: images with unclear boundaries.

From the figure, it can be observed that when segmenting unclear edge regions, LGMANet performs relatively well on most images, with only minor errors on the image in the second row. In contrast, EGE-UNet exhibits significant errors when segmenting the image in the third row, while UNext shows major failures on the second-row image, almost completely failing to segment it.

Figure 10 Segmentation results of different methods on the BUSI dataset: images with unclear boundaries.

From the figure, it can be observed that LGMANet produces relatively stable segmentation results, whereas methods such as EGE-UNet and MedT exhibit significant errors during segmentation.

Figure 11 Segmentation results of different methods on the GlaS dataset: images with region overlap.

From the figure, it can be observed that LGMANet performs well in segmenting the Region Overlap images, with only minor errors in the image of the third row. In contrast, other methods exhibit significant segmentation errors.

Ablation study

Quantitative ablation study

To validate the effectiveness of the innovative modules proposed in this article, we conducted an ablation study on the LGIPBs and the EMRA modules across all four datasets. The quantitative experimental results are shown in Table 7. Specifically, we refer to the model with both the LGIPBs and the EMRA modules removed as the Baseline, the model with only the LGIPBs removed as Baseline+EMRA, the model with only the EMRA modules removed as Baseline+LGIPB, and the original model (LGMANet) as Baseline+EMRA+LGIPB. Additionally, since directly removing the LGIPB would render the network unusable, we replaced it with a standard convolutional layer. For the EMRA module, however, we directly removed it, as this does not affect the usability of the network.

Table 7 Ablation experiment results.

Dataset	Method	Dice	IoU	Recall	Precision	
ISIC2018	Baseline	89.15	80.75	88.91	89.98	
	Baseline+LGIPB	90.06	83.04	90.59	91.07	
	Baseline+EMRA	90.33	82.69	89.87	91.15	
	Baseline+EMRA+LGIPB	91.95	85.28	93.02	91.23	
BUSI	Baseline	72.18	57.28	69.04	78.55	
	Baseline+LGIPB	75.44	61.47	77.11	76.67	
	Baseline+EMRA	73.39	58.76	71.44	77.83	
	Baseline+EMRA+LGIPB	81.93	70.07	85.09	79.45	
CVC-ClinicDB	Baseline	80.99	68.36	79.49	83.46	
	Baseline+LGIPB	85.42	74.98	83.12	88.54	
	Baseline+EMRA	81.82	69.48	81.61	82.51	
	Baseline+EMRA+LGIPB	90.45	82.67	89.70	91.42	
GLaS	Baseline	85.50	74.88	86.93	84.25	
	Baseline+LGIPB	91.71	84.72	92.37	91.14	
	Baseline+EMRA	88.18	78.90	88.60	87.96	
	Baseline+EMRA+LGIPB	94.11	88.90	95.13	93.15	
Note:

Bold values represent the best scores.

The analysis of Table 7 shows that removing any of the modules from LGMANet leads to a decrease in all metrics across all four datasets. This clearly demonstrates that the LGIPBs and the EMRA modules introduced in this article effectively enhance the network’s accuracy in segmenting various types of medical images, including endoscopy, dermoscopy, ultrasound, and H&E-stained histological sections.

Qualitative ablation study

In addition to the quantitative results, we also provide qualitative results to visualize the outcomes of the ablation study, thereby offering a clearer validation of the effectiveness of the proposed modules. As shown in Fig. 12, ablating any of the proposed innovative modules leads to a significant decline in segmentation performance across images from all four datasets. This clearly demonstrates the effectiveness of our proposed modules.

Figure 12 Visualization results of the ablation study.

Here, GT denotes the ground truth, Original Model represents the results produced by the full model, RM-LGIPB refers to the results after removing the LGIPB module, RM-EMRA indicates the results after removing the EMRA module, and RM-All corresponds to the results after removing both LGIPB and EMRA modules.

Conclusion

This article introduces LGMANet, a novel architecture designed for medical image segmentation, with the aim of addressing common challenges such as insufficient extraction of local and global information and inaccurate identification of core features in medical images. To enable effective learning of both local and global information, LGMANet integrates the local-global information processing blocks (LGIPBs). Additionally, to enhance the model’s ability to more accurately select core image features and suppress irrelevant information, the efficient multi-scale reconstruction attention (EMRA) modules are incorporated. Extensive experiments demonstrate that LGMANet outperforms current state-of-the-art methods across various metrics on multiple datasets. The model’s generalization ability and effectiveness have been validated on datasets representing different body parts.

Although LGMANet demonstrates excellent performance across multiple scenarios, certain limitations still exist. First, the model’s performance is constrained on datasets with limited sample sizes, especially when the number of training samples is small. Second, although this study verifies the cross-domain generalization capability of LGMANet across multiple datasets, significant differences among image modalities—such as grayscale distribution, boundary clarity, and structural texture—exist in real-world applications, which may negatively impact model performance. Additionally, external noise in clinical environments may weaken the model’s effectiveness and affect the quality of segmentation results. Finally, due to hardware resource limitations and other factors, this study has not conducted an in-depth investigation of 3D medical image segmentation. However, some researchers have explored this area and proposed efficient models (Hatamizadeh et al., 2021; Ma et al., 2024) such as UNetR++ (Shaker et al., 2024) and Swin-UNetV2 (Liu et al., 2022).

To address these challenges, future work will explore more diverse data augmentation strategies or incorporate transfer learning methods to improve segmentation accuracy in low-data scenarios. We also plan to investigate cross-modal alignment mechanisms or style transfer techniques to enhance the model’s ability to capture semantic consistency across different modalities, thereby improving cross-modal generalization. Additionally, we intend to introduce a discriminator from generative adversarial networks (GANs) into the training process and optimize it adversarially with the generator to enhance the model’s robustness against noise and improve its stability and practicality in real clinical environments. Finally, with adequate hardware support, we will conduct in-depth research on 3D medical image segmentation to develop more accurate and efficient segmentation techniques.

Supplemental Information

Supplemental Information 1 Code.

Additional Information and Declarations

Competing Interests

The authors declare that they have no competing interests.

Author Contributions

Minghui Zhu conceived and designed the experiments, performed the experiments, analyzed the data, performed the computation work, prepared figures and/or tables, authored or reviewed drafts of the article, and approved the final draft.

Dapeng Cheng conceived and designed the experiments, analyzed the data, authored or reviewed drafts of the article, and approved the final draft.

Yanyan Mao conceived and designed the experiments, authored or reviewed drafts of the article, and approved the final draft.

Lu Sun conceived and designed the experiments, authored or reviewed drafts of the article, and approved the final draft.

Wanting Jing conceived and designed the experiments, prepared figures and/or tables, authored or reviewed drafts of the article, and approved the final draft.

Data Availability

The following information was supplied regarding data availability:

The source code is available in the Supplemental File.

The Breast Ultrasound Images Dataset is available at Kaggle: https://www.kaggle.com/datasets/aryashah2k/breast-ultrasound-images-dataset

The CVC-ClinicDB is available at: https://polyp.grand-challenge.org/CVCClinicDB.

The GlaS dataset is available at:

- https://www.kaggle.com/datasets/sani84/glasmiccai2015-gland-segmentation

- https://paperswithcode.com/paper/gland-segmentation-in-colon-histology-images

The ISIC 2018 dataset is available at: https://challenge.isic-archive.com/landing/2018.

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
