# Peer review of "Local-global multi-scale attention network for medical image segmentation"

_PeerJ Computer Science, doi:10.7717/peerj-cs.3033_

## Round 0.1 · original submission · Major Revisions

Dear authors,

You are advised to critically respond to all comments point by point when preparing an updated version of the manuscript and while preparing for the rebuttal letter. Please address all comments/suggestions provided by reviewers, considering that these should be added to the new version of the manuscript.

Kind regards,
PCoelho

Reviewer 1 ·

Basic reporting

This paper introduces LGMANet, a novel medical image segmentation architecture designed to address limitations in existing deep learning methods, such as the inadequate extraction of local and global features and ineffective selection of core features. LGMANet integrates two key components: the Local-Global Information Processing Block (LGIPB), which enhances feature learning through dense connections, and the Efficient Multi-Scale Reconstruction Attention (EMRA) module, which combines attention mechanisms with multi-scale convolutions to improve feature selection and suppress irrelevant information. Experimental results on four benchmark datasets (ISIC2018, CVC-ClinicDB, BUSI, and GLaS) demonstrate that LGMANet achieves superior IoU scores of 85.28%, 82.67%, 70.07%, and 88.90%, respectively, outperforming state-of-the-art methods.

Concerns and Suggestions:
1. The authors claim that current deep learning-based medical image segmentation methods cannot effectively extract both local and global information.
→ Please provide evidence or references to support this claim. Visualizations or comparisons with existing methods would strengthen the argument.
2. Please verify whether the datasets used are outdated. For instance, ISIC2018 may be considered too old to reliably demonstrate the effectiveness of the proposed method.
3. For Equations 8–11, it would improve clarity to present them in a single line if possible.
4. The authors should include additional visualizations, such as latent feature maps, beyond the current prediction results, to better validate the effectiveness of the proposed modules.
5. Missing references:
TP-DRSeg: Improving Diabetic Retinopathy Lesion Segmentation with Explicit Text-Prompts Assisted SAM
SegMamba: Long-range Sequential Modeling Mamba for 3D Medical Image Segmentation
Serp-Mamba: Advancing High-Resolution Retinal Vessel Segmentation with Selective State-Space Model
NestedFormer: Nested Modality-Aware Transformer for Brain Tumor Segmentation
Towards Realistic Semi-Supervised Medical Image Classification
Video-Instrument Synergistic Network for Referring Video Instrument Segmentation in Robotic Surgery

Experimental design

no comment

Validity of the findings

no comment

Cite this review as

·

Basic reporting

1. This paper has some grammar issues, and the description should be improved. For example, line 40 says, "reducing treatment costs. reducing treatment costs." This is ambiguous.
2. They also miss essential segmentation references, such as UNETR++: Delving into Efficient and Accurate 3D Medical Image Segmentation (doi: 10.1109/TMI.2024.3398728).
3. The overall structure of this paper is acceptable.
4. The paper claims that the current SOTA method lacks global and local information and core feature extraction. Is there any reference to support this idea? Or do the authors conduct experiments to support their argument? They didn't provide sufficient evidence to support their hypotheses.
5. The definitions of all terms are good.
6. For section 3.4, since the shifted MLP is not your design, it should not be highlighted in a separate section.

Experimental design

1. This is an original primary research within the Aims and Scope of the journal.
2. The research question is not well defined, and the issues assumed by this paper are not convincing. As far as I know, no references or experiments indicate these issues.
3. The related work is not organized well. The handcrafted feature extraction methods should not be illustrated with excessive spacing. Since deep learning methods for segmentation problems have been extensively studied, they should be organized according to different mechanisms and categories, such as CNN-based and attention-based methods, not by chronological sequence.
4. The ablation study lacks sufficient detail. The paper claims that different modules have been removed in various comparisons, but it does not explain how to remove them, as they are a necessary part of the framework. For example, the LGIPB is a convolutional encoder in the downsampling process; if it is removed, what will be used for encoding? The encoder of the UNet?
5. Why select these four datasets, which only contain lesion data? Why don't you include normal cases? Could you explain the necessity of using lesion data?
6. The metrics are insufficient; Dice and IoU have similar criteria. Typically, segmentation studies use Dice and Hausdorff distance to evaluate the segmentation results.
7. In Table 5, the recall value of EAV-UNet is better than that of the proposed method, but the Dice and IoU values are inferior. Could you explain why this occurs?
8. In Table 5, the parameters of the UNet and the proposed method differ significantly. This appears unreasonable because the proposed method is based on the UNet model and adds additional modules to it. How do you manage to reduce the parameters of your model? Please provide details for clarification.

Validity of the findings

1. The paper didn't compare their method with cutting-edge methods like UNetR++, Swin-UNetR, and others, making it difficult to determine whether this work is meaningful. They also do not explain why they chose 2D segmentation over a 3D method, which typically yields more accurate segmentation results.
2. I think the dataset they used is open-accessed.
3. Conclusions are good.

Additional comments

Everything I want to say is included in the three areas above.

Reviewer 3 ·

Basic reporting

A great Paper! Thank you for sharing your findings!

- The manuscript is clearly written with professional and unambiguous English throughout. The technical terminology and grammar are appropriate for an academic audience.

- The background and motivation for the work are clearly introduced. The related work section is thorough, covering traditional methods, deep learning approaches, and recent advances with appropriate citations.

-The paper conforms well to PeerJ’s structural expectations. The flow from introduction to methodology, experiments, and conclusions is logical and easy to follow.

- Sufficient information about datasets is included. The raw data sources (for example, ISIC2018, BUSI) are public and correctly referenced.


Areas for Improvement:

- The abstract is slightly verbose and could be tightened to focus more succinctly on the key contributions and results. The model performance was mentioned only in the last sentence.

- There is some minor redundancy between the abstract and introduction regarding the challenges and contributions. Consolidating this information may improve clarity.

- While figures are of generally high quality, some could benefit from additional annotations to clarify what each component represents, especially in qualitative comparisons (Figures 4–11), the difference between segmentation output from your model and other models is not clear, I think all segmentation results are very similar to each other

Experimental design

- The research is within the scope of the journal and presents a novel architecture (LGMANet) for medical image segmentation.

- The research questions are clearly defined: improving local-global information extraction and core feature selection.

-The proposed method (LGMANet) incorporates two key innovations—LGIPB and EMRA—which are well-motivated and described both architecturally and mathematically.



In term of improvements:

- I think the pseudo-codes are straight forward, but the mathematical equations require more clarifications, for example a definition of each dimension in B×Cin×H×W should stated.

- some activation functions require more details such as "Prelu(·) refers to a parametric rectified linear activation function that introduces nonlinearity." It would be great if you mention how did you learn the parameters and what were the parameters you used.

Validity of the findings

- The results demonstrate consistent and significant improvements over several state-of-the-art methods across all datasets used.

- Performance metrics are detailed and statistically meaningful, with LGMANet showing gains in Dice, IoU, and Recall on multiple datasets.

- The authors acknowledge limitations—e.g., performance on small datasets and susceptibility to noise and propose thoughtful directions for future work (e.g., GAN-based enhancements, transfer learning).

- It was a great idea to include the ablation study, which is particularly valuable in establishing the contribution of LGIPB and EMRA modules

In term of improvements:

-While statistical improvements are demonstrated, no statistical significance tests (e.g., p-values) are reported. Including such analyses would help validate the robustness of the observed performance differences.

- The generalization claim across modalities is promising, but additional discussion on the diversity of image characteristics across datasets (e.g., modality differences) would better support this claim.

-I think it would be great if you can consider benchmarking against very recent transformer-based models like Swin-UNetV2 (it was built upon the strengths of Swin Transformer and U-Net) or Medical SAM to further contextualize performance gains.

Cite this review as

---

## Round 0.2 · accepted · Accept

Dear authors, we are pleased to verify that you meet the reviewer's valuable feedback to improve your research.

Thank you for considering PeerJ Computer Science and submitting your work.

Kind regards
PCoelho

Reviewer 1 ·

Basic reporting

no comment

Experimental design

no comment

Validity of the findings

no comment

Additional comments

I believe all my concerns have been well addressed.

Cite this review as